# Associations between trajectories of obesity prevalence in English primary school children and the UK soft drinks industry levy: An interrupted time series analysis of surveillance data

**Nina T. Rogers**[1]*, **Steven Cummins**[2], **Hannah Forde**[1,3], **Catrin P. Jones**[1], **Oliver Mytton**[1,4], **Harry Rutter**[5], **Stephen J. Sharp**[1], **Dolly Theis**[1], **Martin White**[1], **Jean Adams**[1]

1 MRC Epidemiology Unit, University of Cambridge School of Clinical Medicine, Institute of Metabolic Science, Cambridge, United Kingdom, 2 Population Health Innovation Lab, Department of Public Health, Environment and Society, London School of Hygiene and Tropical Medicine, London, United Kingdom, 3 Nuffield Department of Primary Care Health Sciences, University of Oxford, Oxford, United Kingdom, 4 Great Ormond Street Institute of Child Health, London, United Kingdom, 5 Department of Social and Policy Sciences, University of Bath, Bath, United Kingdom

* nina.rogers@mrc-epid.cam.ac.uk

**Data Availability Statement:** Data are available from NHS Digital (https://digital.nhs.uk/data-and-

## Abstract

### Background

Sugar-sweetened beverages (SSBs) are the primary source of dietary added sugars in children, with high consumption commonly observed in more deprived areas where obesity prevalence is also highest. Associations between SSB consumption and obesity in children have been widely reported. In March 2016, a two-tier soft drinks industry levy (SDIL) on drinks manufacturers to encourage reformulation of SSBs in the United Kingdom was announced and then implemented in April 2018. We examined trajectories in the prevalence of obesity at ages 4 to 5 years and 10 to 11 years, 19 months after the implementation of SDIL, overall and by sex and deprivation.

### Methods and findings

Data were from the National Child Measurement Programme and included annual repeat cross-sectional measurement of over 1 million children in reception (4 to 5 years old) and year 6 (10 to 11 years old) in state-maintained English primary schools. Interrupted time series (ITS) analysis of monthly obesity prevalence data from September 2013 to November 2019 was used to estimate absolute and relative changes in obesity prevalence compared to a counterfactual (adjusted for temporal variations in obesity prevalence) estimated from the trend prior to SDIL announcement. Differences between observed and counterfactual estimates were examined in November 2019 by age (reception or year 6) and additionally by sex and deprivation quintile. In year 6 girls, there was an overall absolute reduction in obesity prevalence (defined as >95th centile on the UK90 growth charts) of 1.6 percentage

information/publications/statistical/national-child-measurement-programme) for researchers who meet the criteria for access to confidential data.

**Funding:** NTR, OM, MW, and JA were supported by the Medical Research Council (grant Nos MC_UU_00006/7). This project was funded by the NIHR Public Health Research programme (grant Nos 16/49/01 and 16/130/01) to MW. The views expressed are those of the authors and not necessarily those of the National Health Service, the NIHR, or the Department of Health and Social Care, UK. The funders had no role in study design, data collection and analysis, decision to publish, or preparation of the manuscript.

**Competing interests:** I have read the journal's policy and the authors of this manuscript have the following competing interests. JA is an Academic Editor on PLOS Medicine's editorial board.

**Abbreviations:** AIC, Akaike information criterion; BMI, body mass index; CI, confidence interva; GLS, generalised least squares; IMD, index of multiple deprivation; ITS, interrupted time series; NCMP, National Child Measurement Programme; PP, percentage point; SSB, sugar-sweetened beverage; SDIL, soft drinks industry levy.

points (PPs) (95% confidence interval (CI): 1.1, 2.1), with greatest reductions in the two most deprived quintiles (e.g., there was an absolute reduction of 2.4 PP (95% CI: 1.6, 3.2) in prevalence of obesity in the most deprived quintile). In year 6 boys, there was no change in obesity prevalence, except in the least deprived quintile where there was a 1.6-PP (95% CI: 0.7, 2.5) absolute increase. In reception children, relative to the counterfactual, there were no overall changes in obesity prevalence in boys (0.5 PP (95% CI: 1.0, −0.1)) or girls (0.2 PP (95% CI: 0.8, −0.3)). This study is limited by use of index of multiple deprivation of the school attended to assess individual socioeconomic disadvantage. ITS analyses are vulnerable to unidentified cointerventions and time-varying confounding, neither of which we can rule out.

## Conclusions

Our results suggest that the SDIL was associated with decreased prevalence of obesity in year 6 girls, with the greatest differences in those living in the most deprived areas. Additional strategies beyond SSB taxation will be needed to reduce obesity prevalence overall, and particularly in older boys and younger children.

## Trial registration

ISRCTN18042742.

---

## Author summary

### Why was this study done?

- In England, childhood obesity rates are high with around 10% of reception age children (4/5 years) and 20% of children in year 6 (10/11 years) recorded as living with obesity in 2020.

- Children who are obese are more likely to suffer from serious health problems including high blood pressure, type 2 diabetes, and depression in childhood and in later life.

- In March 2016, to tackle childhood obesity, the UK government announced there would be a soft drinks industry levy (SDIL) on manufacturers of soft drinks to incentivize them to reduce the sugar content of drinks.

### What did the researchers do and find?

- We tracked changes in the levels of obesity in children in England from reception (ages 4/5 years) and year 6 (ages 10/11 years) over time between 2014 and 2020. This analysis involved comparing obesity levels 19 months following the SDIL with predicted obesity levels had the SDIL not happened according to gender of the child and school's area level of deprivation.

- The UK SDIL was associated with an 8% relative reduction in obesity levels in girls aged 10/11 years, equivalent to prevention of 5,234 cases of obesity per year in girls aged 10/11 years, alone. Reductions were greatest in girls whose school was in the 40% most deprived areas.

- No associations were found between the SDIL and changes in obesity levels in boys aged 10/11 years or younger children aged 4/5.

**What do these findings mean?**

- Our findings suggest that the UK SDIL led to positive health impacts in the form of reduced obesity levels in girls aged 10/11 years.

- Further strategies are needed to reduce obesity prevalence in primary school children overall, and particularly in older boys and younger children.

## Introduction

There is strong evidence that consumption of sugar-sweetened beverages (SSBs) increases the risk of serious diseases including type 2 diabetes, cardiovascular disease, dental caries, and obesity [1–3]. Children and adolescents in the United Kingdom are particularly high consumers of added sugars [4] with consumption typically peaking at approximately 70 g/day in late adolescence, equivalent to over twice the recommended maximum intake of 30 g [5]. SSBs are the primary source of free sugar in the diets of children and are associated with weight gain, obesity, and fatness in children [6–8]. Demographic patterns of SSB and added sugar consumption mirror each other with highest consumption in older children [5,9], boys [9,10], and children from lower socioeconomic groups [11–13]. Recently born cohorts of children are much more likely to have obesity than children from older cohorts such that 10-year-olds born after the 1980s are 2 to 3 times more likely to develop obesity than those born before the 1980s [14]. The persistence of obesity from childhood into adulthood [15] and its acute and chronic negative physical [16–19] and mental [16,20] health consequences in children has led to governments around the world focusing on preventive strategies to reduce obesity in early life.

The World Health Organization recommends taxes on SSBs to reduce consumption of added sugars to improve health [21]. Over 50 jurisdictions have implemented taxes on soft drinks, although they differ in terms of how much tax is passed through to the consumer, the types of soft drink targeted and the structure of the tax (including banded structure [22] and taxes levied in terms of volume sold [23] or as a proportion of the price [24]). In March 2016, the UK government proposed a number of strategies, including a soft drinks industry levy (SDIL) on manufacturers, importers, and bottlers of SSBs, to reduce prevalence of obesity in childhood [25]. The two-tier SDIL, implemented in April 2018, differed from most other tax structures in that it was designed to incentivise manufacturers to reformulate higher sugar soft drinks to move them to a lower tax tier. Manufacturers and importers were subject to a charge of £0.24/litre on soft drinks containing ≥8 g of sugar per 100 ml, £0.18/litre on soft drinks containing between ≥5 to <8 g of sugar per 100 ml, and no levy on drinks containing <5 g sugar per 100 ml [26]. Levy exempt drinks include milk, milk-based drinks, 100% fruit juice, and powders used to make drinks. As part of the broader health strategy for young people, the UK government indicated they would use revenues raised through the SDIL to fund physical education in schools and breakfast and after-school clubs [27].

Evidence suggests that the UK SDIL led to substantial reformulation of the UK soft drinks market. The percentage of drinks containing >5 g sugar/100 ml fell from 49% to just 15% between September 2015 and February 2019, with reformulation accelerating after announcement of the UK SDIL [28]. Overall, the UK SDIL was associated with a reduction in sugar purchased from soft drinks [29]. While the price of soft drinks increased following implementation of the SDIL, the levy was only partially passed on to the consumer. For example, in drinks containing between ≥5 to <8 g of sugar per 100 ml, approximately one-third of the levy was passed on [28]. A number of modelling studies [30–33] have predicted that the

introduction of SSB taxes would lead to a modest reduction in obesity in children and adults at the population level, but no study to date has used empirical data to examine whether the response of the SSB industry to the UK SDIL was associated with a subsequent change in the prevalence of childhood obesity. A few studies have used empirical data to estimate associations between SSB taxes and weight-related outcomes in children and adolescents and have either shown no overall association [23,34–36] or small to modest associations in specific subgroups such as low-income households [36] children with higher body mass indices (BMIs) [36,37] or in adolescent girls but not boys [38]. Different findings from these discrete studies may be related to use of different outcome measures (in particular, one study relied on subjective measures of self-reported weight [34]), differences in change in SSB prices achieved by taxes (some were associated with small average increases in prices of SSB (<5%) [34,35]) or differences in substitutions to high-calorie untaxed food [23] and drinks [23,35].

In this study, we use cross-sectional data on monthly prevalence of objectively assessed obesity in children when they enter (reception class; ages 4 to 5) and exit (year 6; ages 10 to 11) English primary schools to examine whether 19 months following the implementation of the UK SDIL there were changes in the trajectory of prevalence of obesity (1) overall and (2) by sex and deprivation.

## Methods

The study was registered (ISRCTN18042742) and the study protocol published [39]. This study is reported as per the REporting of studies Conducted using Observational Routinely-collected health Data (RECORD) Statement (S1 Checklist).

### Data source

We used population level data from the National Child Measurement Programme (NCMP). This surveillance programme began in 2006 and measures the height and weight of approximately 1 million children from English state-maintained primary schools in reception (ages 4 to 5 years) and year 6 (ages 10 to 11 years) annually, with the aim of monitoring national rates of overweight and obesity in children. Local authorities oversee the data collection, and letters are sent to the parents of eligible children where they are informed about why the data are collected and how these are stored. There is also an opportunity to opt out of measurement. Approximately 99% of eligible schools (approximately 17,000 schools) take part each year and individual response rates are high with over 90% of eligible pupils taking part [40].

Surveillance data provided by NCMP include prevalence of children with overweight or obesity by school class (reception or year 6), sex (male or female), school year (e.g., 2013/14), month of measurement, and the index of multiple deprivation (IMD) quintile of the location of the primary school that the child attends. The NCMP measures the height and weight of children in England throughout the academic school year (September to July); hence, there was no available data for the month of August when the long summer holiday takes place. IMD scores are commonly used in England as measures of multiple deprivation by considering seven distinct domains including income, employment, education, barriers to housing, health and disability, crime, and living environment [41]. The BMI thresholds used to derive overweight and obesity prevalence values were based on the 85th and 95th centiles, respectively, of a reference sample of measures taken in the UK in 1990 taking account of height, weight, sex, and age, reflecting the definitions used by Public Health England for population surveillance [42]. The study period was initially planned to end 2 years following the implementation of SDIL, but follow-up was curtailed in November 2019

(4 months prior to the proposed end date) to avoid any influence of potential household storing of food and drink in preparation for (i) the UK leaving the European Union (December 2019) and (ii) national lockdown because of the COVID-19 pandemic (March 2020) [43] to avoid contamination with documented changes in weight status occurring in the pandemic [44].

## Statistical analysis

Interrupted time series (ITS) analyses were conducted to assess obesity prevalence in relation to the UK SDIL in children attending primary school reception or year 6 classes, overall and by sex and IMD quintile. The ITS used monthly data from September 2013 (study month 1) until November 2019 (study month 69), including the months of the SDIL announcement (March 2016; study month 29) and implementation (April 2018; study month 52).

Generalised least squares (GLS) models were used. Autocorrelation in the time series was examined visually using plots of autocorrelation and partial autocorrelation and statistically using Durbin–Watson tests; an autocorrelation-moving average (ARIMA) correlation structure was used, with the order (p) and moving average (q) parameters chosen to minimise the Akaike information criterion (AIC) in each model. School holidays are reported to influence weight-related outcomes in school children [45]. To take account of this and other key events in the academic calendar year that might impact weight, we used calendar months as a proxy. Following a standard data-driven approach, to identify which calendar months might predict significant changes in obesity prevalence, we ran a series of GLS models in which a single calendar month was added to the equation. After all, calendar months were tested individually; models were finalised by including all the months that showed significant changes in obesity prevalence. Adding all months as dummy variables was avoided to restrict the number of variables to those that were informative, to reduce error, and to increase the precision of our estimates. The months of September, October, June, and February were significant for reception class children, and September and July were significant for year 6 children. Models for year 6 and reception age children were examined separately because reception age children in England typically start school full time, a few weeks after older children have returned. Model specifications for year 6 and reception class children are included (S1 Text). Counterfactual scenarios were estimated based on pre-announcement trends (S1 Fig). Absolute and relative differences in prevalence of obesity between observed and counterfactual values were estimated at month 69 (November 2019). Confidence intervals were calculated from standard errors estimated using the delta method [46]. All statistical analyses were performed in R version 4.1.0.

## Sensitivity analysis 1: Inclusion of two alternative interruption points

The main analysis included a counterfactual based on the pre-announcement trend (i.e., a scenario where neither the announcement nor implementation happened); however, previous research suggests that reformulation of drinks began some months after the announcement of SDIL but before implementation [28]. Therefore, as well as capturing the earliest possible time when reformulation could come into effect, in sensitivity analyses (S1 Fig), we used two alternative interruption points. First (sensitivity analysis 1a), we used a counterfactual based on the trend from September 2013 to November 2016 (equivalent to 8 months post-announcement and the point at which reformulation increased rapidly) [28]. Second (sensitivity analysis 1b), we used a counterfactual based on the pre-implementation trend, i.e., from September 2013 to April 2018.

### Sensitivity analysis 2: Combining overweight and obesity prevalence

In addition to examining prevalence of obesity, the main analysis was repeated and broadened to examine trajectories of excess weight prevalence, in relation to the SDIL, using monthly measures of overweight in addition to obesity.

## Results

Table 1 summarises the mean obesity prevalence in the study period (i) before the SDIL announcement and (ii) after the SDIL announcement, in primary school children in reception and year 6, overall and by sex and IMD quintile. Highest levels of obesity were observed in the most deprived areas regardless of age and sex; pupils in schools from the most deprived IMD quintiles had nearly twice the prevalence of obesity as those in the least deprived IMD quintiles.

### Changes in obesity prevalence in relation to SDIL

Unless stated otherwise below, all estimates of changes in prevalence of obesity are based on values from November 2019 with respect to the counterfactual scenario of no SDIL announcement or implementation having occurred.

Across all year 6 children, there was a 0.8-percentage point (PP) (95% confidence interval (CI): 0.3, 1.3) absolute reduction or 3.6% (95% CI: 1.2, 5.9) relative reduction in obesity prevalence compared to the counterfactual (see Table 2). Year 6 children in schools from the most deprived IMD quintiles (IMD1 and 2) had the greatest (relative) reductions in obesity prevalence of 4.1% (95% CI: 1.8, 6.3) and 5.5% (95% CI: 3.3, 7.7), respectively; however, large differences between year 6 girls and boys were observed. In year 6 girls, there was an overall relative

**Table 1. Mean obesity prevalence (standard deviation) in the pre- and post-announcement periods of the UK SDIL, by school class, sex, and IMD quintiles.**

| | Mean (standard deviation) obesity prevalence in primary school children in reception[1] and year 6[2] class | | | | | |
| --- | --- | --- | --- | --- | --- | --- |
| | Total population | | Boys | | Girls | |
| | Pre-announcement[3] | Post-announcement[4] | Pre-announcement | Post-announcement | Pre-announcement | Post-announcement |
| School class: Reception[1] | | | | | | |
| All IMD | 9.5(0.9) | 9.8(0.9) | 9.8(1.9) | 10.0(2.2) | 9.0(1.9) | 9.4(2.1) |
| IMD 1 (most deprived) | 11.9(0.6) | 12.5(0.8) | 12.2(0.6) | 12.9(0.8) | 11.5(0.7) | 12.2(1.0) |
| IMD 2 | 10.6(0.8) | 11.0(0.8) | 11.1(0.8) | 11.4(1.0) | 10.1(1.0) | 10.6(0.9) |
| IMD 3 | 9.1(0.7) | 9.5(1.0) | 9.4(0.6) | 9.8(1.0) | 8.8(0.9) | 9.3(1.1) |
| IMD 4 | 8.3(0.9) | 8.5(0.7) | 8.8(1.1) | 8.6(0.7) | 7.8(0.9) | 8.3(1.0) |
| IMD 5 (least deprived) | 7.0(0.8) | 7.1(0.8) | 7.4(0.8) | 7.3(0.8) | 6.8(1.1) | 6.7(0.9) |
| School class: Year 6[2] | | | | | | |
| All IMD | 19.2(0.5) | 20.1(0.6) | 20.8(3.9) | 22.1(4.6) | 17.3(3.8) | 17.9(4.0) |
| IMD 1 | 24.3(1.0) | 26.0(0.7) | 26.1(1.1) | 28.4(1.0) | 22.5 (1.1) | 23.5(0.8) |
| IMD 2 | 21.8(0.7) | 23.1(0.7) | 23.4(0.9) | 25.4(1.1) | 20.1(0.9) | 20.7(0.8) |
| IMD 3 | 19.0(0.6) | 19.7(0.9) | 20.7(0.8) | 21.7(1.3) | 17.2(0.9) | 17.6(0.8) |
| IMD 4 | 16.6(0.7) | 17.2(0.8) | 18.3(0.8) | 19.2(1.0) | 14.8(0.9) | 15.2(0.9) |
| IMD 5 | 13.8(0.7) | 14.2(0.6) | 15.4(1.0) | 15.9(0.9) | 12.2(0.7) | 12.3(0.7) |

[1]Reception class–ages 4/5.

[2]Year 6 class–ages 10/11.

[3]Pre-announcement period = September 2013–March 2016.

[4]Post-announcement period = April 2016–November 2019.

IMD, index of multiple deprivation; SDIL, soft drinks industry levy.

**Table 2. Absolute and relative changes in prevalence of obesity (95% CIs), compared to the counterfactual[1], in reception and year 6 boys and girls, by IMD at 19 months post-implementation of the UK SDIL.**

| | Total population | | Boys | | Girls | |
|---|---|---|---|---|---|---|
| Interruption–SDIL announcement | PP change | Relative change (%) | PP change | Relative change (%) | PP change | Relative change (%) |
| Reception | | | | | | |
| All IMD | 0.3(0.9, −0.3) | 3.0(−3.1, 9.1) | 0.5(1.0, −0.1) | 4.5(−1.0, 10.0) | 0.2(0.8, −0.3) | 2.4(−3.6, 8.4) |
| IMD 1 (most deprived) | −0.5 (0.1, −1.1) | −3.9(−8.4, 0.6) | −0.4(0.2, −0.9) | −2.6(−6.7, 1.4) | −0.6(0.1, −1.2) | −4.3(−9.0, 0.4) |
| IMD 2 | **0.7(1.2, 0.2)** | **6.7(2.0, 11.4)** | **1.2(2.1, 0.4)** | **11.1(3.3, 18.9)** | 0.3(0.9, −0.3) | 2.6(−3.2, 8.4) |
| IMD 3 | **0.9(1.7, 0.2)** | **9.7(1.6, 17.9)** | 0.7(1.7, −0.3) | 7.3(−2.7, 17.4) | **1.2(1.8, 0.5)** | **13.0(5.4, 20.5)** |
| IMD 4 | **0.5(1.0, 0.1)** | **6.3(1.0, 11.6)** | 0.5(1.1, −0.2) | 5.4(−2.2, 12.9) | 0.3(0.6, −0.1) | 3.5(−0.6, 7.6) |
| IMD 5 (least deprived) | **0.6(1.1, 0.1)** | **10.0(2.2, 17.9)** | **0.6(1.1, 0.1)** | **9.7(2.0, 17.4)** | **0.6(1.2, 0.003)** | **10.8(0.1, 21.5)** |
| Year 6 | | | | | | |
| All IMD | **−0.8(−0.3, −1.3)** | **−3.6(−5.9, −1.2)** | −0.04(0.6, −0.6) | −0.2(−2.7, 2.4) | **−1.6 (−1.1, −2.1)** | **−8.0 (−10.5, −5.4)** |
| IMD 1 | **−1.1(−0.5, −1.8)** | **−4.1(−6.3, −1.8)** | 0.2(0.9, −0.5) | 0.6(−1.8, 3.0) | **−2.4(−1.6, −3.2)** | **−9.0(−12.1, −5.9)** |
| IMD 2 | **−1.4(−0.8, −1.9)** | **−5.5(−7.7, −3.3)** | **−0.9(−0.1, −1.7)** | **−3.3(−6.2, −0.4)** | **−2.5(−2.1, −2.9)** | **−11.0(−12.7, −9.2)** |
| IMD 3 | 0.01(0.6, −0.6) | 0.04(−3.0, 3.1) | 1.0(2.4, −0.5) | 4.5(−2.1, 11.1) | −0.5(0.2, −1.2) | −2.8(−6.5, 0.9) |
| IMD 4 | 0.2(0.8, −0.4) | 1.1(−2.1, 4.4) | 0.3(1.0, −0.5) | 1.3(−2.3, 4.8) | 0.2(0.9, −0.5) | 1.2(−3.40, 5.9) |
| IMD 5 | 0.3(0.8, −0.3) | 1.9(−1.8, 5.6) | **1.6(2.5, 0.7)** | **10.1(4.3, 15.9)** | **−0.9(−0.3, −1.5)** | **−7.0(−11.6, −2.3)** |

[1]Estimated from pre-announcement trends.

CI, confidence interval; IMD, index of multiple deprivation; PP, percentage point; SDIL, soft drinks industry levy.

reduction in obesity prevalence of 8.0% (95% CI: 5.4, 10.5). Analysis by IMD revealed greatest reductions in the two most deprived IMD quintiles (1 and 2) of 9.0% (95% CI: 5.9, 12.1) and 11.0% (95% CI: 9.2, 12.7), respectively, where a clear break in trend was observed graphically some months following the SDIL implementation (Fig 1). In year 6 boys, there was no overall change in obesity prevalence and no obvious pattern in changes in prevalence by IMD quintile, although there was a large relative increase in obesity prevalence of 10.1% (95% CI: 4.3, 15.9) in the least deprived IMD quintile and a small reduction in prevalence of obesity in IMD2 of 3.30% (95% CI: 0.4, 6.2) (Fig 2).

In reception children, compared to the counterfactual, there was no absolute change in obesity prevalence overall in girls (0.2 PP (95% CI: 0.8, −0.3)) and boys (0.5 PP (95% CI: 1.0, −0.1)). Examination by IMD and sex showed a consistent increase in prevalence of obesity, compared to the counterfactual, in the least deprived IMD groups in both girls (0.6 PP (95% CI: 1.2, 0.003)) (Fig 3) and boys (0.6 PP (95% CI: 1.1, 0.1)) (Fig 4) in reception class.

When the interruption point was changed to December 2016 (8 months post-SDIL announcement, the point at which reformulation began, sensitivity analysis 1a), changes in obesity prevalence were consistent with the main findings, with reductions in obesity prevalence evident in year 6 children, specifically girls from schools in the most deprived areas (IMD 1 and 2) (S1 Table), and increases in obesity prevalence in year 6 boys from the least deprived areas (IMD 4 and 5). When the interruption point was changed to April 2018 (month of SDIL implementation, sensitivity analysis 1b, S2 Table) findings varied from the main analysis, with an overall absolute increase in the prevalence of obesity in reception age children by 0.7 PP (95% CI: 0.1, 1.3). Compared to the counterfactual, there were few significant changes in obesity prevalence in the different year 6 groups, although reductions (e.g., 3.8% (95% CI 5.7, 2.0) in year 6 girls from IMD 2) and increases (e.g., 3.8% (95% CI 0.2, 7.4) in boys in IMD4) were observed in some groups.

Changes in prevalence of excess weight (overweight or obesity) in relation to the UK SDIL were comparable to the main findings on changes in trends in prevalence of obesity,

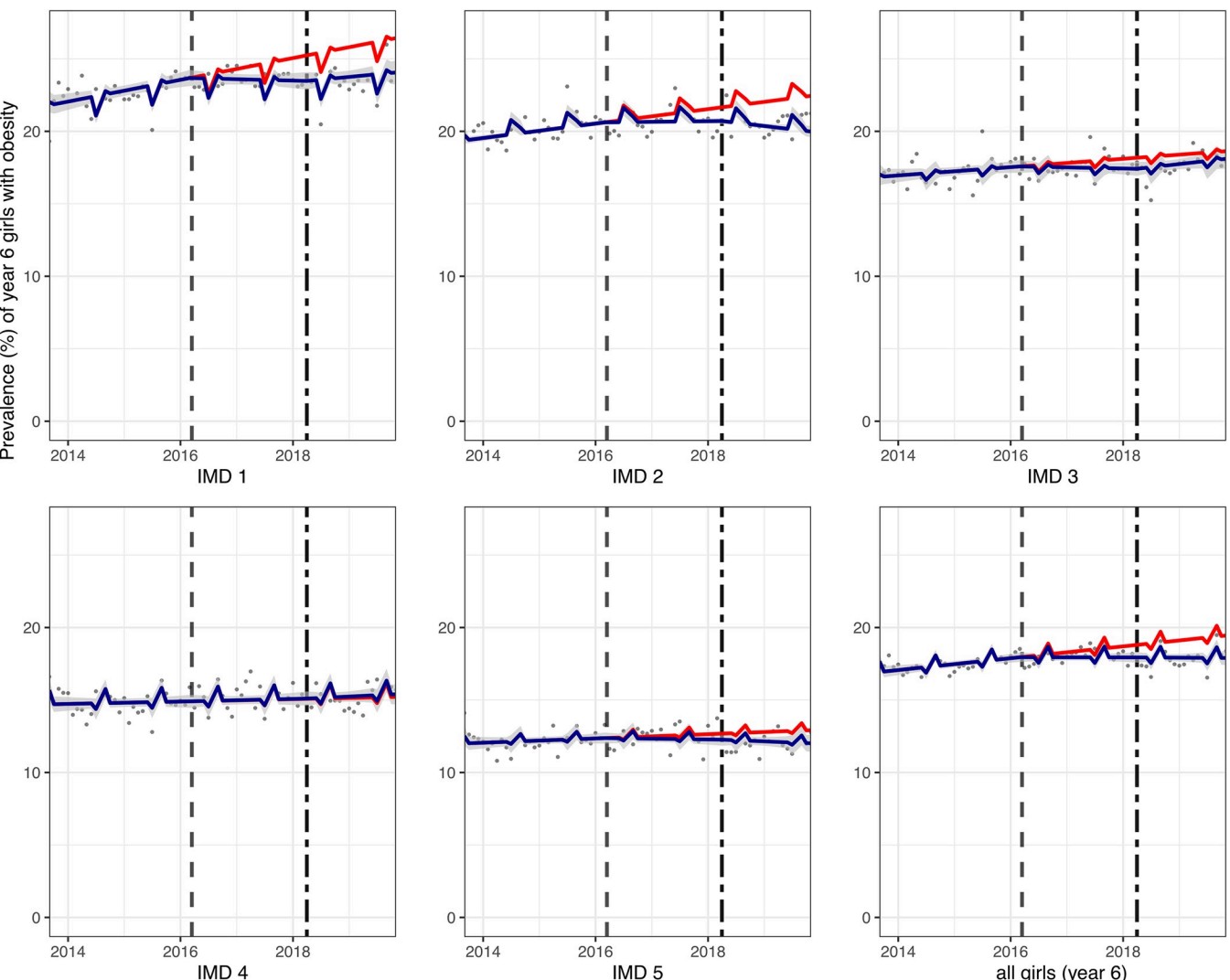

**Fig 1. Prevalence (%) of obesity in year 6 girls (aged 10/11) between September 2013 and November 2019.** Observed and modelled prevalence of obesity is shown by IMD quintile and overall. Dark blue points show observed data and dark blue lines (with grey shadows) shows modelled data (and 95% CIs) of obesity prevalence. The red line indicates the counterfactual line based on the pre-SDIL announcement trend (assuming the announcement and implementation had not occurred). The first and second dashed vertical lines indicate the announcement and implementation of the SDIL, respectively. CI, confidence interval; IMD, index of multiple deprivation; SDIL, soft drinks industry levy.

with greatest reductions in excess weight observed in girls from schools in IMD quintiles 1 and 2 and no change in prevalence of excess weight overall in year 6 boys or reception age children (S3 Table). However, compared to the counterfactual scenario of no announcement or implementation, there was an observed absolute reduction in excess weight of reception age girls from the most deprived IMD (1) of 1.6 PP (95% CI 1.1, 2.1).

## Discussion

### Summary of findings

This is the first study that we are aware of that uses empirical data to examine changes in childhood obesity prevalence in England in relation to the UK SDIL. After accounting for prior trends in obesity, there was a 0.8-PP absolute reduction in year 6 children living

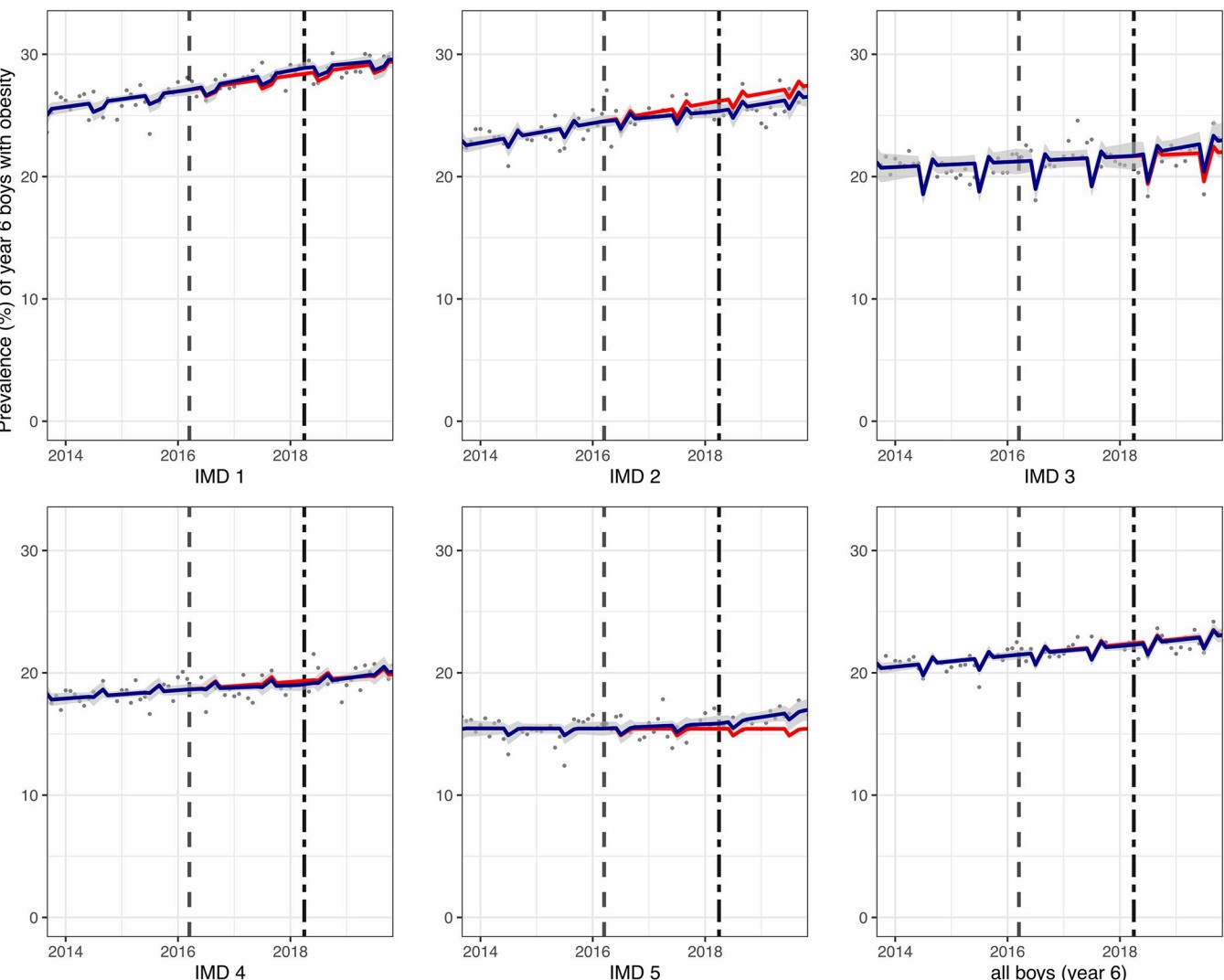

**Fig 2. Prevalence (%) of obesity in year 6 boys (aged 10/11) between September 2013 and November 2019.** Observed and modelled prevalence of obesity is shown by IMD quintile and overall. Dark blue points show observed data and dark blue lines (with grey shadows) shows modelled data (and 95% CIs) of obesity prevalence. The red line indicates the counterfactual line based on the pre-SDIL announcement trend (assuming the announcement and implementation had not occurred. The first and second dashed vertical lines indicate the announcement and implementation of the SDIL, respectively. NB: The scales used in Figs 2–4 differ to maximise resolution of the image. CI, confidence interval; IMD, index of multiple deprivation; SDIL, soft drinks industry levy.

with obesity, 19 months after the implementation of the SDIL. These reductions in year 6 children were predominantly driven by changes in girls, where there was a 1.6-PP absolute or 8.0% relative reduction in obesity prevalence. Assuming, based on our 2019 data, that there are 337,658 year 6 girls in England (of whom 18.4% have obesity), this reduction is equivalent to 5,234 averted cases of obesity in year 6 girls. Relative to the counterfactual, no overall change was observed in year 6 boys. We observed that for year 6 girls, reductions in obesity were greatest in the 40% most deprived IMD areas, with a 2.4-PP absolute or 9.0% relative reduction in the most deprived IMD quintile. Overall, the prevalence of obesity in reception class children was unchanged, compared to the counterfactual.

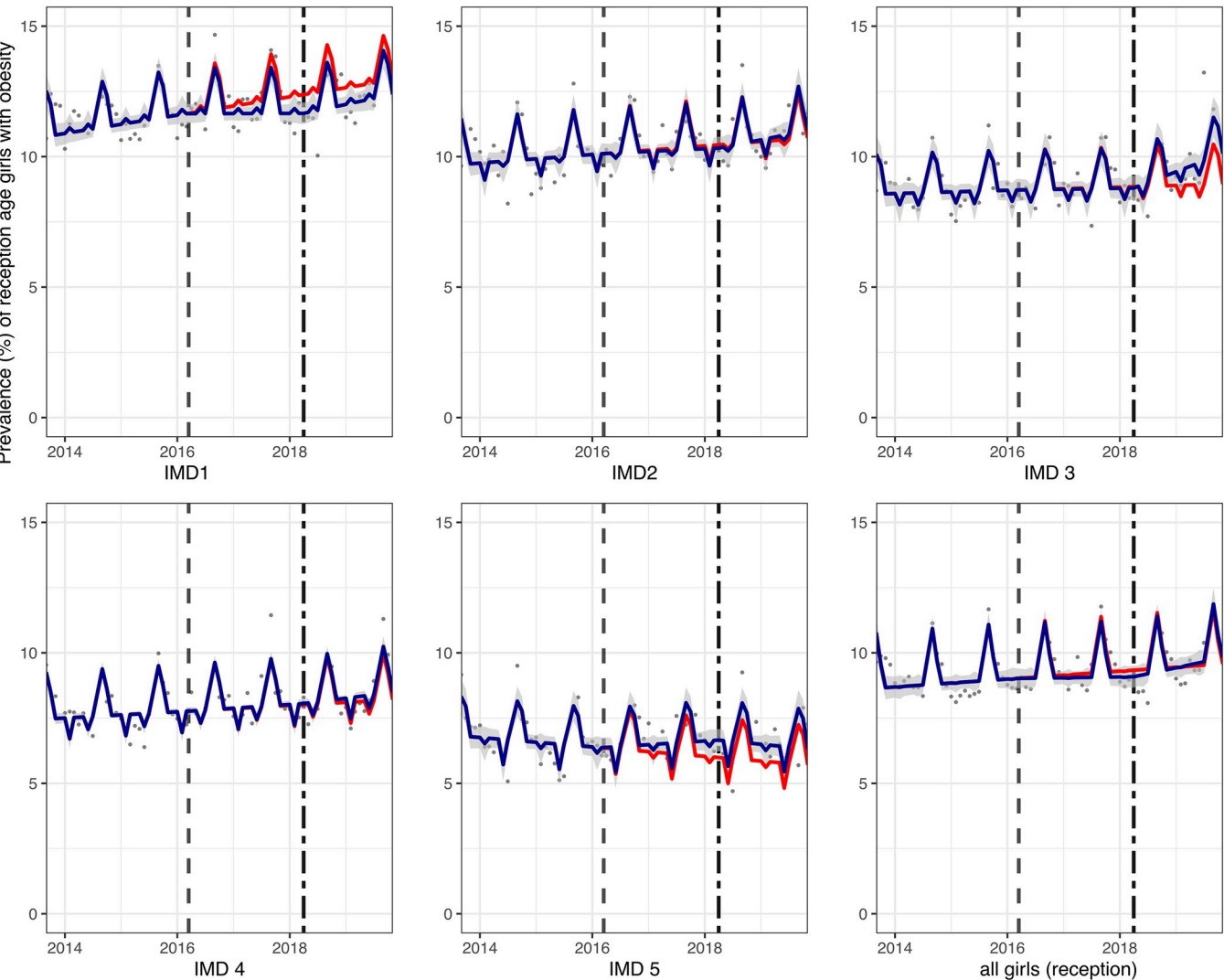

**Fig 3. Prevalence (%) of obesity in girls in reception class (aged 4/5) between September 2013 and November 2019.** Observed and modelled prevalence of obesity is shown by IMD quintile and overall. Dark blue points show observed data and dark blue lines (with grey shadows) shows modelled data (and 95% CIs) of obesity prevalence. The red line indicates the counterfactual line based on the pre-SDIL announcement trends (assuming the announcement and implementation had not occurred). The first and second dashed vertical lines indicate the announcement and implementation of the SDIL, respectively. CI, confidence interval; IMD, index of multiple deprivation; SDIL, soft drinks industry levy.

## Comparison with other studies and implications

In this section, we draw on evidence from other studies and compare our findings with them, while also providing some potential explanations for our results and their implications.

First, our findings are plausible since associations between SSB consumption and risk of obesity are well described in the literature [6–8]. Furthermore, a relationship between the UK SDIL and an overall reduction in sugar purchased from soft drinks across the population has previously been reported [29]. Several modelling studies have also predicted that SSB taxes are likely to be most effective at targeting sugar intake in children and younger adults [47,48].

Second, the magnitude and pattern of associations in our results are consistent with recent findings from Mexico that report a modest reduction in overweight or obesity prevalence in adolescent girls (aged 10 to 18) with a 1.3-PP absolute decrease 2 years after a 10% SSB price

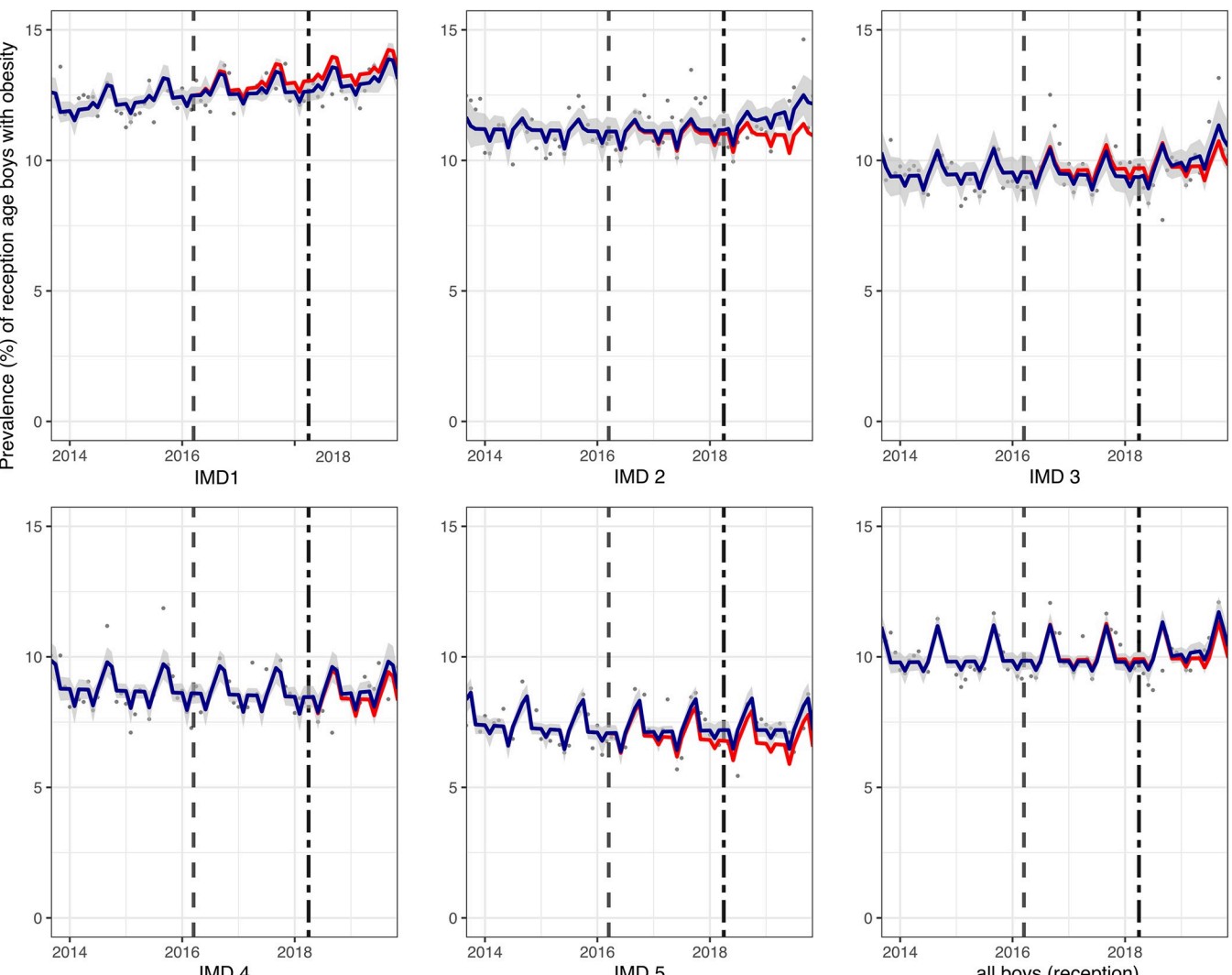

**Fig 4. Prevalence (%) of obesity in boys in reception class (aged 4/5) between September 2013 and November 2019.** Observed and modelled prevalence of obesity is shown by IMD quintile and overall. Dark blue points show observed data and dark blue lines (with grey shadows) shows modelled data (and 95% CIs) of obesity prevalence. The red line indicates the counterfactual line based on the pre-SDIL announcement trends (assuming the announcement and implementation had not occurred). The first and second dashed vertical lines indicate the announcement and implementation of the SDIL, respectively. CI, confidence interval; IMD, index of multiple deprivation; SDIL, soft drinks industry levy.

increase (compared to a 1.6-PP absolute decrease observed in this study in 10- to 11-year-old girls 19 months after the levy was introduced) [38]. Moreover, similar to the findings of this study, no significant reductions in weight-related outcomes were observed in adolescent boys in Mexico. We note, however, that the tax implemented in Mexico is not directly comparable with the UK SDIL; in Mexico; the tax had a different design aimed at increasing the price to consumers resulting in 100% of the SSB tax being passed through to consumers, equating to a 14% increase in prices [49], and, importantly, the tax was included as a wider package of anti-obesity measures, which included charging 8% on high-energy foods [23]. We note the importance of the finding that the tax in Mexico was more effective in girls who were heavier. Similar analysis was not possible here because we only had access to repeated cross-sectional data, which cannot be linked over time.

Third, we found that reductions in obesity in relation to the levy were greatest in children who were older and from the most deprived areas. Previous studies have reported the same children are more likely to be higher consumers of SSBs [5,9,11–13]. This suggests a possible dose–response gradient between consumption levels and effectiveness of the levy in reducing obesity. This also adds to the growing international evidence that SSB taxes may reduce inequalities in diet-related health outcomes. For example, some studies from other countries have shown that lower-income households were more likely to reduce their purchases or intake of sugar from SSBs following introduction of SSB taxes [36,50,51], although this is not always the case [22,52,53].

In this study, we also demonstrate that the UK SDIL is not associated with a change in obesity prevalence in children in the first year of primary school. This result is congruous with findings from a cohort of British children showing that SSB consumption at ages 5 or 7 are not related to adiposity at age 9 years [54]. Added sugars from drinks make up 30% of all added sugars in the diet of young children (aged 1 to 3 years), but this increases to more than 50% by late adolescence [5]. The lower intake of sugars from soft drinks at very young ages may lower the potential of a tax on SSBs, making it harder to observe health effects at the population level. Fruit juices, which are not included in the levy, are thought to contribute similar amounts of sugar in young children's diets as SSBs and may explain why the levy alone is not sufficient to reduce weight-related outcomes in reception age children. In addition to drinks, confectionery, biscuits, desserts, and cakes are also important high-added sugar items, which are regularly consumed by young children and could be a target of additional obesity reduction strategies [5].

While our finding that the SDIL had greater impacts on obesity prevalence in girls than boys is consistent with previous studies [38], it is unclear why this might be the case, especially since boys were higher baseline consumers of SSBs [13]. One explanation is that there were factors (e.g., in food advertising and marketing) at work around the time of the announcement and implementation of the levy that worked against any associations of the SDIL among boys. There is evidence that soft drink manufacturers altered their marketing strategies in different ways in response to the SDIL including repackaging and rebranding products [55]. Numerous studies have found that boys are often exposed to more food advertising content than girls [56–59], both through higher levels of TV viewing [59] and through the way in which adverts are framed. Physical activity is often used to promote junk food, and boys, compared to girls, have been shown to be more likely to believe that energy-dense junk foods depicted in adverts will boost physical performance [56] and thus they are more likely to choose energy-dense, nutrient-poor products following celebrity endorsements. There is also evidence that girls tend to make healthier choices when it comes to diet (e.g., consuming more fruit and vegetables and less energy-dense foods) and other health behaviours (e.g., brushing teeth) [60]. One possibility for the observed differences between boys and girls may be that girls were more responsive to public health signalling arising from discussions around the SDIL or that they were more likely to choose drinks that had been reformulated to contain less sugar following the SDIL announcement.

Even the strongest association of the SDIL among the most levy-responsive groups (e.g., year 6 girls) reflected only a dampening of the rate of increase in obesity prevalence compared to the counterfactual rather than a reversal in trends. This highlights that alongside the SDIL, additional evidence-informed obesity reduction strategies need to be in place to improve weight-related outcomes, especially in boys and younger children, as they enter primary school education.

## Strengths and limitations of the study

This study makes use of a unique and well-powered ongoing nationally representative sample covering over 90% of children aged 4 to 5 and 10 to 11 years in state-run primary schools over

the study period and tracks the prevalence of overweight and obesity in over 1 million school children annually. Obesity prevalence data were based on objective measures of height and weight rather than parental self-report, where there is a tendency to underestimate overweight [61]. The NCMP uses 85th and 95th centiles of the UK1990 growth reference to monitor overweight and obesity in children (accounting for age and sex), respectively [42,62]. However, other cut points are sometimes used [63], and there is some debate over whether this is the best measure of adiposity, particularly in younger children [64].

Parental consent in NCMP involves a selective opt-out, which is designed to increase participation rates. However, it has been suggested that girls with obesity are less likely to participate [65]. This may have led to underestimation of the association between SDIL on obesity prevalence in girls. These effects are, however, likely to be small given that obesity levels in girls have not changed dramatically and participation in the sample overall remained high throughout our study period. Socioeconomic disadvantage was assessed using an area-level indicator (IMD) of the school that each child attended, a less sensitive measure than capturing socioeconomic disadvantage at the household level. However, there is a strong correlation between school-level IMD and the proportion of pupils eligible for free school meals, a measure of the number of children attending a school with a low household income, [66] suggesting that the measure used here is a suitable proxy measure of household deprivation.

Data on time trends of expected childhood weight loss in relation to diet interventions are sparse with studies not monitoring weight-related outcomes with regularity and from early in the intervention. This makes it particularly challenging to estimate how long from the SDIL announcement we would expect to observe changes in obesity prevalence in children. However, there is evidence that changes in energy balance in children can lead to rapid changes in weight loss, for example, seasonal differences in BMI are observed in school children, with weight gain typically occurring during the summer periods especially in children with overweight or obesity [45]. Consistent with these observations, our statistical models and ITS graphs reveal spikes in obesity prevalence in the months following the summer holidays (September in reception and year 6 children, and October in children in reception) and dips in other months (e.g., in June and July) in some subgroups. These require further investigation that could contribute to understanding of seasonal variations in childhood obesity. Furthermore, our ITS graphs reveal that in some groups, there may be continued improvement in the longer term with a widening between counterfactual and observed values in, for example, year 6 girls (IMD 1, 2, and 5).

The ITS approach used modelled counterfactuals on the obesity prevalence trends immediately prior to the SDIL announcement. Given that estimates of the overall difference between observed and counterfactual obesity prevalence can be sensitive to the time points at which the counterfactuals are modelled, as part of a sensitivity analysis, we included two extra interruption points. The first additional interruption was 8 months post-announcement of SDIL, a time when reformulation of SSBs was visibly starting to increase; here we observed very similar findings to the main analysis indicating that they are robust. The second additional interruption was assigned to the date of the SDIL implementation; using this model, we observed fewer significant changes in obesity prevalence compared to the counterfactual (for example, no significant difference was observed in year 6 girls overall). This finding may be explained by the fact that companies had already reformulated most of their products prior to the implementation date and trajectories of obesity prevalence had responded rapidly. Furthermore, examining trajectories of "excess weight" prevalence rather than prevalence of obesity as the outcome of interest led to findings broadly consistent with the main analysis.

## Conclusions

The UK SDIL was proposed by the UK government to tackle childhood obesity. The pattern of findings of this study suggests that the SDIL can contribute to reducing obesity prevalence in older primary school children. The SDIL announcement and implementation was associated with an overall relative decrease in obesity prevalence in year 6 girls aged 10 to 11 years of approximately 8% compared to the counterfactual scenario based on pre-announcement trends. These associations were even greater in girls from schools in the 40% most deprived areas, suggesting the SDIL could help to reduce inequalities in child obesity. Further obesity reduction policies are needed alongside taxes on SSBs to improve and reverse the current obesity prevalence in children.

## Supporting information

**S1 Checklist. RECORD checklist.** RECORD, Reporting of studies Conducted using Observational Routinely-collected Data.
(DOCX)

**S1 Fig. Schematic diagram of the interrupted time series.** Blue solid lines indicate observed data. Dashed red lines represent counterfactuals. Counterfactual for (1) main analysis based on obesity prevalence trends from 09/2013–03/2016; (2)sensitivity analysis (a) based on obesity prevalence trends from 09/2013–12/2016; and (3)sensitivity analysis (b) based on obesity trends from 09/2013–04/2018.
(DOCX)

**S1 Text. Model specifications for children in year 6 and reception class.**
(DOCX)

**S1 Table. Changes in obesity prevalence compared to a counterfactual scenario based on trends prior to 8 months post-announcement.** Absolute and relative changes in prevalence of obesity (95% CIs), compared to a counterfactual scenario[1] based on trends prior to 8 months post-announcement, overall and by IMD in reception and year 6 children, 19 months post-implementation of UK SDIL. CI, confidence interval; IMD, index of multiple deprivation; SDIL, soft drinks industry levy.
(DOCX)

**S2 Table. Changes in obesity prevalence compared to a counterfactual scenario based on trends prior to the SDIL implementation.** Absolute and relative changes in prevalence of obesity (95% CIs), compared to a counterfactual scenario[1] based on pre-SDIL implementation trends, overall and by IMD in reception and year 6 children, 19 months post-implementation of UK SDIL. CI, confidence interval; IMD, index of multiple deprivation; SDIL, soft drinks industry levy.
(DOCX)

**S3 Table. Changes in excess weight prevalence compared to a counterfactual scenario based on trends prior to the SDIL announcement.** Absolute and relative changes in prevalence of excess weight (overweight or obesity) and 95% CIs, compared to a counterfactual scenario[1], based on pre-SDIL announcement trends, overall and by IMD in reception and year 6 children, 19 months post-implementation of UK SDIL. CI, confidence interval; IMD, index of multiple deprivation; SDIL, soft drinks industry levy.
(DOCX)

## Author Contributions

**Conceptualization:** Steven Cummins, Oliver Mytton, Harry Rutter, Stephen J. Sharp, Martin White, Jean Adams.

**Data curation:** Nina T. Rogers.

**Formal analysis:** Nina T. Rogers.

**Funding acquisition:** Steven Cummins, Oliver Mytton, Harry Rutter, Martin White, Jean Adams.

**Investigation:** Nina T. Rogers, Jean Adams.

**Methodology:** Nina T. Rogers, Stephen J. Sharp.

**Project administration:** Catrin P. Jones, Jean Adams.

**Supervision:** Jean Adams.

**Validation:** Nina T. Rogers, Jean Adams.

**Writing – original draft:** Nina T. Rogers, Steven Cummins, Oliver Mytton, Harry Rutter, Stephen J. Sharp, Dolly Theis, Martin White, Jean Adams.

**Writing – review & editing:** Nina T. Rogers, Steven Cummins, Hannah Forde, Catrin P. Jones, Oliver Mytton, Harry Rutter, Stephen J. Sharp, Dolly Theis, Martin White, Jean Adams.

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
