## [Editor Report · Decision Letter 0]

21 Sep 2022

Dear Dr Rogers, 

Thank you for submitting your manuscript entitled "Associations between trajectories of obesity prevalence in English primary school children and the UK soft drink industry levy: an interrupted time series analysis of surveillance data" for consideration by PLOS Medicine.

Your manuscript has now been evaluated by the PLOS Medicine editorial staff and I am writing to let you know that we would like to send your submission out for external peer review.

Please re-submit your manuscript within two working days, i.e. by Sep 23 2022 11:59PM.

Kind regards,

Beryne Odeny

PLOS Medicine

---

## [Decision Letter · Decision Letter 1]

3 Nov 2022

Dear Dr. Rogers,

Thank you very much for submitting your manuscript "Associations between trajectories of obesity prevalence in English primary school children and the UK soft drink industry levy: an interrupted time series analysis of surveillance data" (PMEDICINE-D-22-03100R1) for consideration at PLOS Medicine. 

Your paper was evaluated by an associate editor and discussed among all the editors here. It was also discussed with an academic editor with relevant expertise, and sent to independent reviewers, including a statistical reviewer. The reviews are appended at the bottom of this email and any accompanying reviewer attachments can be seen via the link below:

[LINK]

In light of these reviews, I am afraid that we will not be able to accept the manuscript for publication in the journal in its current form, but we would like to consider a revised version that addresses the reviewers' and editors' comments. Obviously we cannot make any decision about publication until we have seen the revised manuscript and your response, and we plan to seek re-review by one or more of the reviewers. 

We hope to receive your revised manuscript by Nov 24 2022 11:59PM. Please email us (plosmedicine@plos.org) if you have any questions or concerns.

We look forward to receiving your revised manuscript. 

Sincerely,

Callam Davidson, 

PLOS Medicine

plosmedicine.org

Please add this statement to the manuscript's Competing Interests: "JA is an Academic Editor on PLOS Medicine's editorial board."

Please ensure that all numbers presented in the abstract are present and identical to numbers presented in the main manuscript text.

In the last sentence of the Abstract Methods and Findings section, please describe the main limitation(s) of the study's methodology.

Please include continuous line numbering throughout your manuscript to facilitate future reviews.

Please ensure that the study is reported according to the RECORD guideline, and include the completed RECORD checklist as Supporting Information. Please add the following statement, or similar, to the Methods: "This study is reported as per the REporting of studies Conducted using Observational Routinely-collected health Data (RECORD) Statement (S1 Checklist)."

The RECORD guideline can be found here: https://www.equator-network.org/reporting-guidelines/record/

Did your study have a prospective protocol or analysis plan? Please state this (either way) early in the Methods section.

Please present and organize the Discussion as follows: a short, clear summary of the article's findings; what the study adds to existing research and where and why the results may differ from previous research; strengths and limitations of the study; implications and next steps for research, clinical practice, and/or public policy; one-paragraph conclusion. The vast majority of this content is already present so it is largely a matter of restructuring rather than revising. 

Your study is observational and therefore causality cannot be inferred. Please remove language that implies causality, such as ‘effects’ (Conclusions section). Refer to associations instead.

Please provide date accessed for References labelled [Internet].

In Figures 1-4, please show the axis beginning at zero. If this is not possible, please show a break in the axis. Please also ensure that the y axis is identical for all figures to facilitate comparison.

Comments from the reviewers:

Reviewer #1: See attachment

Michael Dewey

Reviewer #2: Overview and general recommendation: 

This study estimates trajectories in the prevalence of obesity for children at ages 4-5 years and 10-11 years 19 months after the implementation of the sugar-sweetened beverages (SSBs) tax in the UK. It describes these trajectories overall, by sex and deprivation using an interrupted time series approach or ITSA. It is one of the first to examine changes in unhealthy weight among children in relation to the SSBs tax. Except for one study in Mexico, previous evidence was largely based on microsimulations or was linking relatively small changes in SSBs prices to health outcomes. Overall, this paper is clearly written, uses a valid empirical approach and research design given the setting, novel data, and presents and interprets the results well. Below are some comments that I hope will help improve it. 

1. I suggest describing SSBs tax in more detail. While the tax is briefly described in "levels", it would be helpful to have a better understanding what it means in terms of %. How does it compare to the Mexican SSBs tax? And to sales SSBs taxes in the US? How is it similar/different? Why was it designed as a function of sugar content? Was it levied on manufacturers? What was the pass-through to consumer prices or is there another mechanism at play? Are there any previous studies estimating SSBs demand elasticities (overall, by sex and deprivation) in the UK? All these things will help the reader understand better why SSBs tax might be working and how. 

2. To this end, the Introduction Is heavily focused on describing historic obesity trends in the UK. I suggest adding information on sugar as well as SSBs intake because that is essentially what SSBs tax is targeting. What is the consumption among children of interest overall, by sex and deprivation? You describe obesity rates, but what is important to understand are baselines of the mechanism through which SSBs tax is likely to operate. 

3. Briefly explain what the evidence on tax & reformulation says exactly, so the reader doesn't have to go to reference [20] to find out. 

4. Statements referring to the existing evidence (and therefore the gap that this study is filling) are not fully correct. The study published in JAMA Pediatrics that studies unhealthy weight changes among adolescents post SSBs tax in Mexico does not use subjective measures, but objectively measured clinical data. Also, in Mexico, SSBs price changes post tax varied, but on average went up about 10 %. Therefore, your rationale that evidence across studies is mixed cannot be that data relies on self-reported weight and that minimal price increases in prices of SSBs are below 5% - those studies mostly study sales SSBs taxes in the US. So, while that may be the case in most previous work studies, it is not true for the studies you are describing just above that explanation. While I understand what you are trying to say, I suggest rewriting the paragraph for correctness. Also, I suggest to more clearly explain how these papers are different from this study in particular, not only how they are different between each other. For instance, you don't link SSBs price changes to health, but do a before/after SSBs tax comparison - using a very novel and well-powered data tracking children's weight over time, which is quite hard to come by and unique, so I suggest stressing that. I also suggest you also look at the paper by Aguilar et al. (2021) for before/after tax comparison in the context of Mexico - I did not see this study on the reference list. You can discuss (here or in other sections) differences in approaches between studies (e.g., regression discontinuity design vs ITSA). 

5. I suggest including calendar month indicators instead of including only selected months by hand. Let the data decide what is needed.

6. It is not clear at what level did you cluster your standard errors. I suggest clustering them at the school level, as you have more than 17000 schools and enough students within each. 

7. Discussion of the absolute results is confusing - you explain absolute differences in %? I believe you should state absolute changes in percentage points not percent. 

8. Were differences in IMD quintiles significantly different from each other? It does not seem to be the case from looking at the confidence intervals, but please, test this formally and report whether that is the case. You can achieve this by including an interaction term in one regression - from what I understand, you estimate each regression model separately for each subgroup. Make sure to explain this clearly in the methods section as well. 

9. Why did obesity prevalence increase among reception children? Report results with numbers for that too to better understand the magnitude and statistical significance of the effects. If magnitudes are small, fine, if not, these results deserve a detailed discussion, too. 

10. You carefully apply several sensitivity analyses and mention your limitation, including the potential issue of selection bias. Can you check whether selection bias is present in the data? Provide results in the appendix. For instance, do you see smaller opting out pre- post-tax in poorer areas? How would this bias your results given that these are the groups responding the most to the tax? 

11. I think comparison of your results to those in Mexico are interesting (e.g., 1.3pp change) - however, it is important to understand how SSBs taxes compare as well. That is why including more details on this and other SSBs taxes in the Introduction (or a bigger discussion in the Discussion) will be very helpful. Also, study in Mexico finds larger effects for heavier girls at baseline - did you check for that in your study? Given that the literature has previously reported higher health elasticities for heavier individuals I suggest you check that in your data as well (overall and by sex). It could also help explaining why your estimates are larger in more deprived areas where obesity seems to be the highest. 

12. This study does not provide causal estimates, so I suggest that you do not refer to them as such. There is no control group (besides the projected counterfactual trend), and this is not a clinical trial. Just because something was reported previously, it does not make it causal - especially if previous literature is heavily observational. Labeling your estimates associations is perfectly fine - nothing is wrong with that, if they are robust. However, study study's research design does not allow us to call them causal. 

13. In the Discussion, you mention that you "…found that reductions in sugar intake in relation to the levy were greatest in children who were likely to be higher consumers of SSBs…" This is confusing - nowhere is this mentioned in the data, analyses or results. Is that from another paper of yours? I suggest rewriting so that it is clear you did not analyze diet data in this study. 

14. Perhaps you want to include a brief discussion that these are short term effects and may increase over time as sugar may be addictive long term. So, in that sense, your estimates are a lower bound. 

15. I like the figures - the break in trend is visible. I suggest you stress the break in trend more in your results discussion as this supports their robustness. I also strongly suggest writing out an equation/regression model you estimate in the appendix, and describe a coefficient of interest, along with controls included etc… Though quite straightforward, I think it will help to understand your methodology better. 

16. As mentioned above, in tables, include interactions to test for difference across subgroups. Otherwise, it is hard to say whether one group had a bigger effect than the other. 

Reviewer #3: This is a well thought-out and well written piece of work. It provides a valuable evaluation of an important public health strategy, in a population with fairly high intakes of SSBs (particularly the year 6 children), providing a good rationale for the analysis. The methods are well described and sound, and you have included sensitivity analyses with various interruption points to account for varied implementation of the SDIL. Incorporating sex and deprivation into the analyses was a useful addition and has allowed for a more thorough understanding of the impact of SDIL. The discussion draws reasonable conclusions and considers the findings in relation to existing work, including regions with a similar strategy in place e.g. Mexico. You have considered key limitations of the work and have tempered your findings by drawing attention to the relatively small, albeit significant, impact of the strategy and the need for additional strategies to tackle the obesity crisis in children. I have suggested some minor additions to the write-up below, where I would have liked some clarification or some consideration of these points. 

Statistical analysis: 

1. Interrupted time series (ITS) analyses were conducted to assess obesity prevalence in relation to the UK SDIL in children attending primary school reception or year 6 classes, overall and by sex and index of multiple deprivation (IMD) quintile - please confirm here that the analyses related to deprivation was by school IMD rather than pupil-level IMD

2. models included adjustment for months that showed significant changes in obesity prevalence (September, October, June and February for reception class and September and July for year 6) - was this for every year within the period? (Are you assuming that these significant changes occur/are the same every year or did you only select the months and years where significant changes were observed?)

Discussion:

1. Are the study team aware of any: 

* pre-emptive reformulation within major drinks brand prior to announcement, that may have provided an additional interruption point, in addition to the two already explored?

* any other obesity prevention strategies over this time period, that correlate with the interruption points, that may have impacted on overweight/obesity trends, e.g. introduction of revised school food standards, roll out of universal infant free school meals? Have these other strategies been considered in your thinking around the attribution of these changes in obesity to the SDIL?

[LINK]

---

## [Decision Letter · Decision Letter 2]

13 Dec 2022

Dear Dr. Rogers,

Thank you very much for re-submitting your manuscript "Associations between trajectories of obesity prevalence in English primary school children and the UK soft drinks industry levy: an interrupted time series analysis of surveillance data" (PMEDICINE-D-22-03100R2) for review by PLOS Medicine.

I have discussed the paper with my colleagues and the academic editor and it was also seen again by three reviewers. I am pleased to say that provided the remaining editorial and production issues are dealt with we are planning to accept the paper for publication in the journal.

[LINK]

We look forward to receiving the revised manuscript by Dec 20 2022 11:59PM.   

Sincerely,

Callam Davidson, 

Associate Editor 

PLOS Medicine

plosmedicine.org

Comments from the Academic Editor:

I am in agreement with Reviewer 1 that the authors ought to have added month dummies and I am unclear as to why they chose not to. At a minimum, the authors need to justify their approach versus all-inclusive month dummies in the paper. 

Requests from Editors:

Please define abbreviations in your Figure legends and also include the age of children in year 6/reception for the benefit of readers unfamiliar with the English school system.

Comments from Reviewers:

Reviewer #1: The authors have addressed mu points but I still feel there is a remaining issue. The data-driven fitting of month dummies seems to me to be undesirable. Deciding on the model in the light of the data means that the usual statements about confidence intervals and p-values become unreliable. It would be simpler just to add in every month.

Michael Dewey

Reviewer #2: Thank you for carefully revising the manuscript. I have two additional comments:

1. formal testing between income subgroups: just because something is monotonically increasing, it does not mean the effects between subgroups are statistically significantly different from zero. If you describe results separately (i.e., not something is bigger than the other), not testing the difference is fine. If, however, you would like to conclude that associations between the tax and weight changes in lower income groups are larger than associations observed in higher income groups, a formal test is, I believe, necessary. Looking at the table 2, confidence intervals seem to be overlapping and it is not obvious to me why a monotonic relationship alone would imply significant differences between groups; so I recommend an empirical test. If you don't apply it, however, I recommend making sure that results are interpreted within each income groups separately (i.e., without between groups comparison). 

2. I understand you do not claim this study's results are causal. However, I recommend that you do not motivate the section "Comparison with other studies and implications" with a sentence on causality either. Your association may be robust, but it is not causal, no matter how in line it is with other correlational studies. Your study is valuable on its own; no need to overstretch its findings. 

Reviewer #3: Thank you for responding to these comments. I am satisfied that the response has dealt with my concerns and I have no further issues to raise.

[LINK]

---

## [Editor Report · Decision Letter 3]

21 Dec 2022

Dear Dr Rogers, 

On behalf of my colleagues and the Academic Editor, Professor Barry Popkin, I am pleased to inform you that we have agreed to publish your manuscript "Associations between trajectories of obesity prevalence in English primary school children and the UK soft drinks industry levy: an interrupted time series analysis of surveillance data" (PMEDICINE-D-22-03100R3) in PLOS Medicine.

PRESS

Sincerely, 

Callam Davidson 

Associate Editor 

PLOS Medicine